# Post-healing follow-up study of patients in remission for diabetic foot ulcers Pied-REM study

Marie Bouly[1], Francois-Xavier Laborne[2], Caroline Tourte[2], Elodie Henry[2], Alfred Penfornis[1,3], Dured Dardari [1,4] *

1 Diabetology Department, Centre Hopitalier Sud Francilien, Corbeil-Essonnes, France, 2 Unité de Recherche Clinique Centre Hospitalier Sud Francilien, Corbeil-Essonnes, France, 3 Paris-Sud Medical School, Paris-Saclay University, Gif-Sur-Yvette, France, 4 LBEPS, Univ Evry, IRBA, Université Paris Saclay, Evry, France

* dured.dardari@gmail.com

## Abstract

The rate of recurrence for diabetic foot ulcer (DFU) is 50% at 2 years Armstrong DG, 2017. International recommendations call for regular monitoring to prevent DFU recurrence. We aim to investigate the relation between post-healing follow-up and recurrence rates. The study will begin in November 2021 and end in April 2022; final study results are scheduled for December 2022. The purpose of the study is to evaluate the benefit of the multidisciplinary follow-up of healed DFU patients at the rate of two annual consultations and its impact on foot wound recurrence.

## 1. Introduction

Diabetic foot ulcers (DFU) are a major health problem throughout the world. The rate of DFU recurrence is 40% in the first year after healing, 50% at 2 years, and 65% at 5 years [1]. The mortality of patients with a history of DFU is 2.5 times greater than those without this history [2]. The main causes of DFU recurrence are inadequate footwear, lack of pedicure care, and lack of selfcare [3]. The International Working Group on the Diabetic Foot (IWGDF) recommends specific podiatric management to prevent the risk of recurrence of trophic disorders; this management must be reassessed or repeated every 1 to 3 months if necessary [4]. At the Centre Hospitalier Sud Francilien, France, the diabetic foot outpatient unit created a specific consultation for healed DFU patients in 2016. Consultations for the management of diabetic foot in remission assess the recommended items in any patient at the rate of two visits per year. In the literature, there is no evidence for preventing the recurrence of foot wounds. Nevertheless, we believe that post-wound follow-up reduces the risk of recurrence.

The study was approved by Centre hospitalier Sud Francilien hospital etichal committee on May 18, 2021 with apouval number _ 26.07.15.38991

## 2. Material and methods

The study protocol was approved by ethics committee of the Centre Hospitalier Sud Francilien and registered in the protocol registration system as NCT04892771

---

---

**Funding:** The author(s) received no specific funding for this work.

**Competing interests:** he authors have declared that no competing interests exist.

## 2.1. Description of study protocol

Pied-REM is a retrospective observational study. Comprehensive information about the study was provided to each patient in a document printed for this purpose. Since 2016, the Diabetology Unit in the Centre Hospitalier Sud Francilien has offered a specialized bi-annual consultation to all patients in remission for DFU. This study will assess the relevance of this human and material investment in the context of a multidisciplinary follow-up approach (a team made up of a podiatrist, podo-orthositis, orthoprotesitis, a nurse specializing in the management of diabetic foot and a diabetologist)The 2-year follow-up period will allow for a sufficient number of patients to be included: the first patients will be monitored from 2017 to 2019 and the last ones from 2019 to 2021Two groups of patients will be defined according to the number of consultations held over a 2-year period. Patients who complete at least two annual visits for 2 years or two sessions over 2 years will be included in the experimental group. Patients who do not attend this minimum number of visits will be included in the control group.

## 2.2. Study objectives

The main objective of the study was to evaluate the impact of the multidisciplinary follow-up of healed DFU patients on wound recurrence. Our secondary objectives were to evaluate the requirements for hospitalization due to recurrence, the pedicure follow-up of patients, and their use of suitable footwear.

## 2.3. Participants

The inclusion criteria are patients aged 18 years and over with type 1 or 2 diabetes who receive follow-up care at the Centre Hospitalier Sud-Francilien for healed DFU. Patients must freely give informed consent to participate in the study. Exclusion criteria are patient refusal.

All study patients have a high podiatric risk.

## 2.4. Data collection

The data will be collected in patient files in the computer database of the diabetology department over a 2-year period. The collected data are the following: verification of inclusion criteria, age, sex, type and duration of diabetes, presence of arterial disease, diabetic neuropathy, history of amputation, and glycosylated hemoglobin.

During the 2-year observation period from the date of inclusion, we will collect the following information: number of follow-up consultations over 2 years from inclusion, recurrence of foot wounds, hospitalization for foot wounds, pedicure follow-up, and use of suitable footwear.

## 2.5. Statistical methods

Regarding the sample size, to demonstrate a difference of 25% to 40% between two groups, with a risk of first species of 5%, a power of 80%, and continuity correction, 158 patients are required. We estimate that 200 diabetic patients are currently being monitored at the hospital for multidisciplinary "foot remission" consultations, which should largely be able to enlist the required number of patients.

Categorical variables will be presented as numbers and percentages and compared using the exact Fisher test or chi2 test. Quantitative variables will be presented as medians and interquartile ranges or as means and standard deviations and compared using the t-test or Wilcoxon test according to their distribution. Recurrence, hospitalization, pedicure follow-up, and use of adapted footwear will be compared between the two groups using the exact Fisher test.

The time to recurrence in each group will be measured using the Kaplan-Meier estimator and compared between the two groups using a log-rank test.

All tests will be carried out bilaterally, with a risk of first species alpha set at 5%. Statistical analysis will be conducted with R software [5].

## 3. Discussion

The Pied-REM study begins on November 2021, with the duration of inclusion set at 1 year. Results will be published around December 2022.

The rate of recurrence of foot wounds remains very high despite improved management [1]. The recurrence of wounds has been evaluated using different approaches: preventive surgical interventions [6], insoles with integrated sensors [7], and follow-up with podiatrists [8] or nurses [9]. Temperature monitoring also helps to limit recurrence [10], as does the use of therapeutic shoes and treatment adherence [11]. In the literature, only one study incorporates all the IWGDF recommendations [4]. Another study from Lithuania showed that multidisciplinary follow-up reduces the risk of recurrence of foot wounds [12]. However, as this study is quite dated (1999) and has a relatively small population size (56 patients), the subject deserves a new evaluation.

The wound recurrence rate is the primary assessment outcome in our study, which was previously evaluated in a meta-analysis [13]. A study on the economic impact of hospitalization costs related to DFU recurrence shows that patients are less often amputated but more frequently hospitalized [14]. Podiatric monitoring recommended by IWGDF can reduce the number of recurrences [8]. Wearing suitable footwear is also part of the recommendations, as the use of customized footwear reduces recurrence rates [3]. We therefore want to show that customized and therapeutic shoes reduce recurrence. The studied population corresponds to all the patients followed for a DFU. As it has been demonstrated that an initial ulceration is highly predictive of recurrence, these patients should be closely monitored [15].

This monocentric retrospective observational study will allow us to justify our approach used for the past 5 years, in which remission foot consultations are offered to all patients with healed DFU. The two study groups include the experimental group that attends all the recommended consultations (two per year) and the control group that does not. The study will compare the rate of wound recurrence between the two groups. The expected findings of this study are a decrease in the number of recurrences in patients with optimal post-healing follow-up. If the results are verified, this will confirm our hypothesis and our approach based on the IWGDF recommendations [4].

The limitation of this study is the methodology, as a retrospective observational study has a very low level of evidence, since it is not randomized and its two groups can have heterogeneous sizes. In this type of study, randomization seems unethical, since it would result in a loss of chance for patients in the control group who would not benefit from post-healing follow-up.

## Supporting information

**S1 Checklist. SPIRIT 2013 checklist: Recommended items to address in a clinical trial protocol and related documents**\*.
(DOC)

**S1 Fig. Post-healing follow-up study of patients in remission for diabetic foot ulcers Pied-REM study.**
(DOC)

**S1 File.**
(DOCX)

**S2 File.**
(PDF)

## Author Contributions

**Conceptualization:** Marie Bouly, Francois-Xavier Laborne, Dured Dardari.

**Data curation:** Marie Bouly, Francois-Xavier Laborne, Caroline Tourte, Alfred Penfornis.

**Formal analysis:** Francois-Xavier Laborne, Caroline Tourte, Elodie Henry.

**Investigation:** Marie Bouly, Dured Dardari.

**Methodology:** Marie Bouly, Dured Dardari.

**Project administration:** Marie Bouly, Dured Dardari.

**Validation:** Marie Bouly.

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
