## [Decision Letter · Decision Letter 0]

7 Feb 2022

PONE-D-21-28523POST-HEALING FOLLOW-UP STUDY OF PATIENTS IN REMISSION FOR DIABETIC FOOT ULCERS Pied-REM studyPLOS ONE

Dear Dr. dardari,

Thank you for submitting your manuscript to PLOS ONE. After careful consideration, we feel that it has merit but does not fully meet PLOS ONE’s publication criteria as it currently stands. Therefore, we invite you to submit a revised version of the manuscript that addresses the points raised during the review process.

We look forward to receiving your revised manuscript.

Kind regards,

Federico Ferrari

Academic Editor

PLOS ONE

Journal Requirements:

2. Please include “Protocol” in the manuscript  title.

Reviewers' comments:

Reviewer's Responses to Questions

**Comments to the Author**

1. Does the manuscript provide a valid rationale for the proposed study, with clearly identified and justified research questions?

Reviewer #1: Yes

Reviewer #2: Yes

2. Is the protocol technically sound and planned in a manner that will lead to a meaningful outcome and allow testing the stated hypotheses?

Reviewer #1: Partly

Reviewer #2: Yes

3. Is the methodology feasible and described in sufficient detail to allow the work to be replicable?

Reviewer #1: Yes

Reviewer #2: Yes

4. Have the authors described where all data underlying the findings will be made available when the study is complete?

Reviewer #1: No

Reviewer #2: Yes

5. Is the manuscript presented in an intelligible fashion and written in standard English?

Reviewer #1: Yes

Reviewer #2: Yes

6. Review Comments to the Author

You may also provide optional suggestions and comments to authors that they might find helpful in planning their study.

Reviewer #1: Congratulation of well done effort in your clinical practive. DFU is a complicated and frustrating issue to many patients and physicians and hopefully you will have meaningful results to help the medical community.

I do have few points:

- Correct the world “Mai” to “May” in the Introduction

- "The study protocol was approved by" was mentioned twice in same page, in the Introduction and in the Description of study protocol. Possibly to remove it from the Introduction.

- In the Study description, it was mentioned that the study is a retrospective observational and the part mentioned in Data collection about collecting patients files and reviewing their cases seems appropriate with the definition of this study. But also mentioned in the Data collection that there will be a 2-year observation period from the date of inclusion with different variables intended to be measured which seems to be a different period than the initial one described. So, will there be a prospective observational part in this study? If yes, then it needs to be mentioned clearly in the Description of study protocol.

- Mentioned in the Statistical methods that there are two groups, experimental and control based on the number of visits, this study protocol needs to be mentioned in the Description of study protocol, not in the Statistical methods.

- It seems from the Statistical method that the measured outcome is based on having or not having an intervention which is the number of visits or sessions over 2 years between the two groups (experimental and control). That has not been clearly defined in the Study objectives.

- There are study protocol description information in the discussion that needs to placed in the material and methods and better to review and place needed information in appropriate place. For example: “The 2-year follow-up period will allow for a sufficient number of patients to be included: the first patients will be monitored from 2017 to 2019 and the last ones from 2019 to 2021”, this sentence belongs to Material and methods.

Reviewer #2: In the part of the abstract, the author mentioned that ,The rate of recurrence for diabetic foot ulcer (DFU) is 50% at 2 years. For this statement need to add the reference.( what is the source of information).

In the part of Introduction, study objective part the author mention the word the multidisciplinary

follow-up of healed DFU patients, but did not explore what author mean by multidisciplinary.

In the part of Participants- the author did not mention the total number of sample size, Gender of the Participants, and average age of the participants.

The study only based on Quantitative methodology. If the authors conduct the study by mixed methodology (quantitative and qualitative) its would more efficient to explore the socio-economic reasons/ factors behind the yearly follow-up for the experimental group and explore the socio-economic reasons/ factors behind are barriers are works for the yearly follow-up for the control group.

7. PLOS authors have the option to publish the peer review history of their article (what does this mean?). If published, this will include your full peer review and any attached files.

Reviewer #1: No

Reviewer #2: No

---

## [Author Response · Author response to Decision Letter 0]

2 Mar 2022

Response to Reviewers

Dear editor and reviewers:

Many thanks for your comments and for the interest you have given to our manuscript please find the answers to your remarks

Reviewer #1

1 - Correct the world “Mai” to “May” in the Introduction: Done ( we are very sorry for this mistake)

2- "The study protocol was approved by" was mentioned twice in same page, in the Introduction and in the Description of study protocol. Possibly to remove it from the Introduction : modification done 

3- In the Study description, it was mentioned that the study is a retrospective observational and the part mentioned in Data collection about collecting patients files and reviewing their cases seems appropriate with the definition of this study. But also mentioned in the Data collection that there will be a 2-year observation period from the date of inclusion with different variables intended to be measured which seems to be a different period than the initial one described. So, will there be a prospective observational part in this study? If yes, then it needs to be mentioned clearly in the Description of study protocol. 

thank you very much for your comment, the study is completely retrospective with an observation period of 2 years post healing of a foot in a living diabetic patient, there is no prospective component

4- Mentioned in the Statistical methods that there are two groups, experimental and control based on the number of visits, this study protocol needs to be mentioned in the Description of study protocol, not in the Statistical methods.

Modification done 

5- - It seems from the Statistical method that the measured outcome is based on having or not having an intervention which is the number of visits or sessions over 2 years between the two groups (experimental and control). That has not been clearly defined in the Study objectives.

Modification done 

6-There are study protocol description information in the discussion that needs to placed in the material and methods and better to review and place needed information in appropriate place. For example: “The 2-year follow-up period will allow for a sufficient number of patients to be included: the first patients will be monitored from 2017 to 2019 and the last ones from 2019 to 2021”, this sentence belongs to Material and methods

Modification done 

Reviewer # 2

1-In the part of the abstract, the author mentioned that ,The rate of recurrence for diabetic foot ulcer (DFU) is 50% at 2 years. For this statement need to add the reference.( what is the source of information).

Reference added 

2-In the part of Introduction, study objective part the author mention the word the multidisciplinary

follow-up of healed DFU patients, but did not explore what author mean by multidisciplinary

Thank you very much for your comment, the explanation has been added

3-In the part of Participants- the author did not mention the total number of sample size, Gender of the Participants, and average age of the participants.

thank you very much for your comment, at the current state it is proposed that the study design the results of patient demographics will be proposed in the final publication, thank you for your understanding

4-The study only based on Quantitative methodology. If the authors conduct the study by mixed methodology (quantitative and qualitative) its would more efficient to explore the socio-economic reasons/ factors behind the yearly follow-up for the experimental group and explore the socio-economic reasons/ factors behind are barriers are works for the yearly follow-up for the control group.

thank you very much for this comment, we are going to make an amendment to study the socio-economic reasons associated with the recurrence of wounds

---

## [Decision Letter · Decision Letter 1]

26 Apr 2022

POST-HEALING FOLLOW-UP STUDY OF PATIENTS IN REMISSION FOR DIABETIC FOOT ULCERS Pied-REM study

PONE-D-21-28523R1

Dear Dr. dardari,

We’re pleased to inform you that your manuscript has been judged scientifically suitable for publication and will be formally accepted for publication once it meets all outstanding technical requirements.

Kind regards,

Federico Ferrari

Academic Editor

PLOS ONE

Additional Editor Comments (optional):

Reviewers' comments:

Reviewer's Responses to Questions

**Comments to the Author**

1. Does the manuscript provide a valid rationale for the proposed study, with clearly identified and justified research questions?

Reviewer #1: Yes

Reviewer #2: Yes

2. Is the protocol technically sound and planned in a manner that will lead to a meaningful outcome and allow testing the stated hypotheses?

Reviewer #1: Partly

Reviewer #2: Yes

3. Is the methodology feasible and described in sufficient detail to allow the work to be replicable?

Reviewer #1: Yes

Reviewer #2: Yes

4. Have the authors described where all data underlying the findings will be made available when the study is complete?

Reviewer #1: No

Reviewer #2: Yes

5. Is the manuscript presented in an intelligible fashion and written in standard English?

Reviewer #1: Yes

Reviewer #2: Yes

6. Review Comments to the Author

You may also provide optional suggestions and comments to authors that they might find helpful in planning their study.

Reviewer #1: My comments were responded approximately. There are non additional comments. Best of luck to authors.

Reviewer #2: I read the manuscript overall. The authors meet all of my quires and now it seems sound good. Hopefully, the authors reviews it again and submit it.

7. PLOS authors have the option to publish the peer review history of their article (what does this mean?). If published, this will include your full peer review and any attached files.

Reviewer #1: No

Reviewer #2: **Yes: **S M Raduan Hossin

---

## [Editor Report · Acceptance letter]

10 May 2022

PONE-D-21-28523R1 

POST-HEALING FOLLOW-UP STUDY OF PATIENTS IN REMISSION FOR DIABETIC FOOT ULCERS
Pied-REM study 

Dear Dr. dardari:

I'm pleased to inform you that your manuscript has been deemed suitable for publication in PLOS ONE. Congratulations! Your manuscript is now with our production department. 

Kind regards, 

on behalf of

Dr. Federico Ferrari 

Academic Editor

PLOS ONE